# Nicotinamide Riboside Supplementation to Suckling Male Mice Improves Lipid and Energy Metabolism in Skeletal Muscle and Liver in Adulthood

**DOI:** 10.3390/nu14112259

**Published:** 2022-05-28

**Authors:** Alba Serrano, Andreu Palou, M. Luisa Bonet, Joan Ribot

**Affiliations:** 1Laboratori de Biologia Molecular, Nutrició i Biotecnologia (LBNB), Nutrigenomics, Biomarkers and Risk Evaluation Group, Universitat de les Illes Balears, 07122 Palma, Spain; albaserranobengoechea@gmail.com (A.S.); andreu.palou@uib.es (A.P.); joan.ribot@uib.es (J.R.); 2CIBER de Fisiopatología de la Obesidad y Nutrición (CIBERobn), 07122 Palma, Spain; 3Institut d’Investigació Sanitària Illes Balears (IdISBa), 07120 Palma, Spain

**Keywords:** early-life nutrition, B3 vitamin, liver metabolism, metabolic programming, muscle metabolism

## Abstract

Nicotinamide riboside, an NAD^+^ precursor, has been attracting a lot of attention in recent years due to its potential benefits against multiple metabolic complications and age-related disorders related to NAD^+^ decline in tissues. The metabolic programming activity of NR supplementation in early-life stages is much less known. Here, we studied the long-term programming effects of mild NR supplementation during the suckling period on lipid and oxidative metabolism in skeletal muscle and liver tissues using an animal model. Suckling male mice received a daily oral dose of NR or vehicle (water) from day 2 to 20 of age, were weaned at day 21 onto a chow diet, and at day 90 were distributed to either a high-fat diet (HFD) or a normal-fat diet for 10 weeks. Compared to controls, NR-treated mice were protected against HFD-induced triacylglycerol accumulation in skeletal muscle and displayed lower triacylglycerol levels and steatosis degree in the liver and distinct capacities for fat oxidation and decreased lipogenesis in both tissues, paralleling signs of enhanced sirtuin 1 and AMP-dependent protein kinase signaling. These pre-clinical findings suggest that mild NR supplementation in early postnatal life beneficially impacts lipid and energy metabolism in skeletal muscle and liver in adulthood, serving as a potential preventive strategy against obesity-related disorders characterized by ectopic lipid accumulation.

## 1. Introduction

Nicotinamide riboside (NR) is a form of vitamin B3 naturally contained in foods that functions as a precursor of NAD^+^ in mammals [1]. NAD^+^ is an essential redox cofactor for fuel oxidation and a rate-limiting substrate for sirtuins enzymes (SIRTs), which sense NAD^+^ levels to regulate multiple cellular processes, including energy metabolism. The NAD metabolome, particularly levels of NAD^+^ and/or NADP^+^ and NADPH, declines in target tissues in humans and rodents suffering from different forms of metabolic stress, such as aging, type 2 diabetes, and fatty liver (reviewed in [2,3,4]). Dietary NR supplementation appears to be beneficial in these contexts through replenishment of the NAD metabolome [2,3,4]. Among the NAD^+^ precursors, NR is of particular interest owing to its potency in vivo and lack of adverse clinical effects [2,4].

Supplementation with NR protects adult mice on obesogenic diets from weight gain and the development of insulin resistance and fatty liver [5,6,7]. NR supplementation favors liver health as it attenuates hepatic fat accumulation, fibrosis, low-grade inflammation, and liver damage in dietary animal models of hepatosteatosis [7,8], a genetic model of type 2 diabetes (KK/HIJ mice) [9], and in alcohol-fed mice [10,11]. Benefits of NR supplementation have also been described in animal models of muscle disorders, such as models of mitochondrial myopathies [12,13] and of muscular dystrophy characterized by muscle wasting [14,15]. Furthermore, NR treatment improves muscle quality and physical performance in middle-aged mice [16] and preserves muscle mass during weight loss in obese rats under calorie restriction [17]. The positive outcomes of these and other pre-clinical studies are prompting human trials assaying the impact of NR supplementation on a variety of metabolic and exercise-related end-points in healthy adults, old persons, and persons with obesity, with disparate results so far [18,19,20,21,22,23]. Mechanistically, the benefits of NR have been linked to increased mitochondria abundance and oxidative function in key metabolic tissues through NAD^+^ activation of SIRT1 [5,7,10,12,13], an enhancement of the differentiating capacity of adult progenitor cells [16], and effects on the gut microbiota [11,24]. 

NR is present in milk [1,25,26], which opens the door to programming effects on newborn development and future metabolism. Only recently, these aspects have begun to be explored [27,28,29]. Maternal NR supplementation (3 g/kg) during the lactation period enhances in mice maternal postpartum weight loss, milk production, and nursing behavior, resulting in weanlings with larger size and improved glycemic control, and it confers lasting advantages regarding behavioral/cognitive skills, physical performance, and body composition of offspring in adulthood [29]. Effects on offspring were largely attributed to increased transmission of macronutrients, micronutrients, and brain-derived neurotrophic factor (BDNF) into milk [29]. In a different approach, we have shown that direct, oral NR supplementation to suckling mice improves metabolic responses to high-fat diet (HFD) feeding in adulthood selectively in the male offspring, in which increases in lipolysis and the plasma leptin-to-adiponectin ratio following HFD feeding were blunted [27]. These beneficial effects are associated with a greater HFD-induced oxidative and thermogenic transcriptional response in brown and white adipose tissues (BAT and WAT) [27] and with changes in DNA methylation marks in brown fat-related genes in subcutaneous WAT [30]. Furthermore, early-life NR supplementation increased the commitment of progenitor cells resident in WAT of young male mice (but not young females) towards the beige (versus white) adipogenesis transcriptional program [28]. Collectively, these results indicated that NR supplementation during the suckling period programs for increased WAT energy metabolism in adulthood in male offspring. Whether early-life NR treatment could also influence skeletal muscle (SM) and liver metabolism features and responses to HFD feeding in the long-term in these animals remained to be studied. These aspects are addressed here.

## 2. Materials and Methods

### 2.1. Animal Experiment

The animal study protocols were reviewed and approved by the Bioethical Committee of the University of the Balearic Islands (Ref. 3513), and the handling of laboratory animals was according to international standards. The study design has been described in detail previously [27]. In brief, suckling NMRI mice in size-adjusted litters (12 pups/litter) received daily from postnatal day 2 to 20, with the aid of a pipette, a small volume (10–15 μL) of vehicle (water, control mice) or NR solution (Chemical Point, Desisenhofen, Germany) providing approximately 15-fold the total vitamin B3 ingested daily from maternal milk [27]. Weaning was on day 21, on a standard chow diet. The male offspring were used. From day 75 to day 90, the animals were habituated to a purified normal-fat diet (NFD, 10% energy as fat; Research Diets, New Brunswick, USA, ref. D12450J). On day 90, half of the animals in each treatment group (vehicle and NR) began receiving a purified high-fat diet (HFD, 45% energy as fat, mainly lard; Research Diets, ref. D12451) while the other half remained on the NFD, until euthanization on day 164 of age, making a total of 4 study groups (*n* = 5–6 animals/group, from 4–5 different litters: mice in each litter were split among the 4 study groups). The animals were euthanized by decapitation, under fed conditions, within the first 2 h of the light cycle. SM (*gastrocnemius*) and liver were dissected in their entirety, snap-frozen in liquid nitrogen, and stored at −80 °C until processing). Housing conditions were 2–3 animals per cage at 22 °C, with a 12 h light-dark cycle (lights on at 08:00) and ad libitum access to food and water. An independent experiment was conducted in which control and NR-treated male mice were euthanized at age 35 days (*n* = 5–6 animals/group). Early-life NR supplementation and the subsequent study protocol followed in this work were conducted in parallel with the study of long-term effects of early-life resveratrol supplementation [31] so that both works shared the control groups (i.e., the non-supplemented siblings studied after the NFD/HFD challenge in adulthood). Except for *gastrocnemius* muscle and liver weight of adult animals, other biometric parameters, as well as plasma and adipose tissue parameters of animal cohorts used in this work, have been reported elsewhere [27,28] (see also Appendix A).

### 2.2. RNA Isolation and Gene Expression Analysis

Total RNA was extracted from tissues using Tripure Reagent (Roche, Barcelona, Spain) following the supplier’s instructions. The RNA was purified with sodium acetate, quantified using a Nanodrop ND 1000 spectrophotometer (Nano-Drop Technologies Inc., Wilmington, NC, USA), and checked for integrity by means of agarose gel electrophoresis. Reverse transcription, PCR amplification of cDNAs of interest, and data analysis were conducted as previously described [32]. Sequences of primers used (from Sigma-Aldrich, St. Louis, MO, USA) are available upon request. Guanosine diphosphate dissociation inhibitor 1 (*Gdi1*), glyceraldehyde-3-phosphate dehydrogenase (*Gapdh*)*,* or ribosomal protein, large, P0 (*Rplp0*) were used as reference genes for data normalization.

### 2.3. Mitochondrial DNA Content

Tissue total DNA was isolated with the DNeasy Blood & Tissue kit (Qiagen, Austin, TX, USA). qPCR quantification of a mitochondrial fragment DNA and a single-copy nuclear gene (beta-2-microglobulin) served to calculate the mitochondrial DNA to nuclear DNA ratio (mtDNA/nDNA), as previously described [33].

### 2.4. Lipid Extraction and Triacylglycerol Determination

Tissue lipids were extracted from 100 mg liver and 150 mg SM tissue, as previously described [34]. Triacylglycerol levels in the lipid extracts were analyzed using the Serum Triglyceride Determination Kit (Sigma-Aldrich). 

### 2.5. Immunoblotting Analysis

Protein levels of total adenosine monophosphate-activated protein kinase (AMPK), phosphorylated AMPK (p-AMPK), and phosphorylated acetyl-CoA carboxylase (p-ACC) in SM were analyzed by immunoblotting as previously described [31], using the Odyssey Western blot kits (Li-Cor Biosciences, Lincoln, NE, USA). The signals of every protein of interest were normalized to the signal of heat shock protein 90. 

### 2.6. Histological Analysis

Neutral lipid droplets in SM samples were detected by Oil Red O (ORO) staining of sections from frozen tissue, according to [35]. Freshly collected liver samples from 4 mice were fixed in 4% paraformaldehyde in 0.1 M sodium phosphate buffer, pH 7.4, overnight at 4 °C, dehydrated in a graded series of ethanol, and cleared in xylene before being embedded in paraffin blocks. Then, 5-micrometer-thick sections were cut with a microtome, mounted on slides, and stained with hematoxylin/eosin for light microscopy. Steatosis in histological liver sections was classified into grades according to Brunt et al. [36]. Grade 0 was assigned when no fat vacuoles were observed and grades 1, 2, and 3, when fat vacuoles were observed in less than 33%, between 33 to 66%, and more than 66% of hepatocytes, respectively.

### 2.7. Statistical Analysis

Data are expressed as mean ± SEM. Student’s *t*-test was used for single comparisons and two-way ANOVA to analyze the interaction of early-life treatment with the type of diet in adulthood (NFD or HFD). Statistical analyses were performed with IBM SPSS Statistics for Windows, version 27.0 (IBM Corp., Armonk, NY, USA). The threshold of significance was set at *p* < 0.05. Trends are highlighted when 0.05 < *p* < 0.1.

## 3. Results

### 3.1. Early-Life NR Treatment Protected against HFD-Induced Triacylglycerol Accumulation in SM in Adulthood 

To assess the long-term effects of early-life NR treatment on muscle features, we began by considering muscle mass and lipid content in control and treated adult mice on an NFD and after a 10-week HFD feeding period (Figure 1)*. Gastrocnemius* muscle weight in adulthood was not affected by early-life NR treatment when expressed either as absolute value (mg) or relative to body weight (as %) (Figure 1A,B). The relative *gastrocnemius* muscle weight tended to be decreased by HFD feeding, irrespective of the neonatal NR treatment (*p* = 0.059 for diet effect, two-way ANOVA) (Figure 1B). Muscle triacylglycerol content, as determined by biochemical analyses (including both intramyocellular and extramyocellular triacylglycerol), was robustly increased after 10 weeks of HFD in the control mice but not the NR mice, who had lower content than controls after HFD (Figure 1C). Oil Red O staining of frozen muscle sections confirmed the presence of more intramyocellular neutral lipid droplets in SM of HFD-fed as compared to NFD-fed control mice and of fewer lipid droplets in the NR mice as compared to the control mice after HFD (Figure 1D). Thus, NR supplementation in the early stages of postnatal life suppressed the accumulation of triacylglycerols in SM induced by HFD feeding later in life. 

### 3.2. Early-Life NR Treatment Impacted Lipid Metabolism and Mitochondria Pathways in SM in Adulthood 

Expression levels of genes related to fatty acid oxidation (FAO)/lipolysis (Figure 2A), lipogenesis/lipid droplet dynamics (Figure 2B), and mitochondria biogenesis/function (Figure 2C) were assessed in SM of control and NR mice after the 10-week NFD/HFD challenge. In good concordance with the muscle triacylglycerol content results, gene expression of *Cpt1b*—Encoding carnitine palmitoyl transferase 1 (CPT1), isoform b, the rate-limiting enzyme in muscle mitochondrial FAO—was increased by HFD feeding in the NR mice, but not the control mice (significant interactive treatment × diet effect, two-way ANOVA). Moreover, gene expression levels of *Acacb*—Encoding acetyl-CoA carboxylase (ACC, isoform b) producing the CPT1 inhibitor malonyl-CoA [37,38]—were lower in SM of NR mice compared with controls under HFD (*p* < 0.05, *t*-Student). Gene expression levels of *Ucp3*—encoding uncoupling protein 3, whose expression in SM highly correlates with the tissue FAO rate [21,22]—were greater in the NR mice, as indicated by two-way ANOVA. We also assessed the mRNA levels in SM of *Ppara* and *Ppard*, coding for peroxisome proliferator-activated receptor (PPAR) transcription factors involved in the transcriptional control of FAO, namely PPARɑ and PPARδ, respectively. NR mice displayed lower *Ppara* mRNA levels than controls under NFD, yet an HFD-induced decrease in *Ppara* mRNA levels present in the control mice was absent in the NR mice, resulting in a significant interactive treatment × diet effect (two-way ANOVA). Likewise, HFD feeding resulted in decreased *Ppard* mRNA levels in SM in control mice (*p* < 0.05, *t*-Student) but not in NR mice. Muscle gene expressions of *Acaca* (encoding ACC, isoform a)*, Ucp2* (encoding uncoupling protein 2), and *Pnpla2* (encoding lipolytic adipose triglyceride lipase) were not affected by early-life NR treatment or the fat content of the diet (Figure 2A). 

Observed differences in the expression levels in muscle of genes related to lipogenesis and lipid droplet metabolism were also in good concordance with the muscle triacylglycerol content results. Gene expression of *Pparg*, encoding PPARγ, was significantly decreased in SM of NR mice, and that of *Plin2*, encoding Perilipin2, trended lower as well in the NR mice (*p* = 0.057 for the NR effect, two-way ANOVA) (Figure 2B). PPARγ is a lipogenic transcription factor whose activity in SM has been related to fat infiltration in the tissue [39], and there is evidence that Perilipin2 participates in intramyocellular lipids synthesis and lipid droplet growth [40]. Interestingly, expression in SM of *Plin5*, encoding Perilipin5, was markedly and significantly induced after HFD feeding selectively in the NR mice but not the control mice. This is remarkable, considering that Perilipin5 anchors mitochondria to the lipid droplet membrane and that coupling via Perilipin5 augments mitochondria respiratory capacity [41].

The mtDNA/nDNA was used as a surrogate index of mitochondria content. HFD feeding increased this ratio in SM of control mice, in keeping with previous reports [42,43] but not further in the NR mice, which tended to have a higher mtDNA/nDNA ratio than controls under basal (NFD) conditions (*p*= 0.088, *t*-Student) (Figure 2D). Differences among groups in the SM mtDNA/nDNA ratio grossly mirrored differences in expression in SM of *Mfn2* (encoding mitofusin 2) and *Ppargc1a* (encoding PPAR gamma coactivator 1 alpha) (Figure 2C). Strikingly, SM *Tfam* mRNA levels were lower in the NR mice than in the control mice under NFD, yet they were increased following HFD feeding in the NR mice only, resulting in a significant interactive treatment × diet effect (two-way ANOVA). Gene expression in SM of other genes related to mitochondria biogenesis and function analyzed (*Tfb2m, Ppargc1b, Gabpa*) was similar in all experimental groups (Figure 2D).

### 3.3. Early-Life NR Treatment Up-regulated AMPK and SIRT1 in SM in Adulthood 

SIRT1 is a key mediator of the NR effects on energy metabolism [5], and SIRT1 and AMPK can positively feedback each other’s activity to activate FAO and mitochondrial biogenesis [44,45]. Interestingly, HFD feeding led to increased muscle *Sirt1* mRNA levels selectively in the NR mice (Figure 3A), who also had a greater phospho-AMPK to AMPK ratio in muscle than control mice, especially following HFD feeding (Figure 3B). Additionally, muscle mRNA levels of *Prkaa1,* encoding one of two known AMPK catalytic subunit isoforms, were up-regulated in the NR mice compared with controls under NFD (*p* = 0.053, *t*-Student), albeit they were increased upon HFD in the control mice only (Figure 3A). These results are compatible with a relative up-regulation of the SIRT1/AMPK axis in SM of adult NR mice. However, known downstream transcriptional SIRT1 responses [46], namely *Gadd45a* and *Sod2* expression, were not affected by the early-life NR treatment or the fat content of the diet (Figure 3A), and the same was true for the levels in muscle of the AMPK product phospho-ACC (Figure 3C). 

### 3.4. Early-Life NR Treatment Resulted in Decreased Liver Triacylglycerol Content and Modified Lipid Metabolism Capacities in Liver in Adulthood 

Liver weight was increased by HFD feeding irrespective of NR treatment (Figure 4A), while the liver index (percentage of liver weight/body weight) was unaffected (Figure 4B). Liver triacylglycerol content, as measured through biochemical analysis, was unaffected by the HFD regimen imposed (Figure 4C), yet it was significantly lower in the NR mice than in the control mice both after NFD and HFD, as indicated by two-way ANOVA. Liver steatosis as a histological diagnosis was analyzed and semi-quantified (Figure 4D). Histological analyses of liver sections revealed an increased degree of steatosis in the HFD-fed animals as compared to the NFD-fed animals and a lower degree of steatosis in the NR mice as compared to the control mice, both after NFD and HFD feeding (Figure 4D,E). 

Hepatic mRNA expression levels in the fed state of sets of genes related to FAO and lipolysis (*Cpt1a*, *Ucp2*, *Ppara*, *Pnpla2*), lipogenesis and lipid droplets (*Fasn*, *Acaca*, *Pparg*, *Srebf1*, *Mlxipl, Plin2*, *Plin5*), and the status of SIRT1 and AMPK pathways (*Sirt1*, *Gadd45a*, *Sod2*, *Prkaa1*, *Prkaa2*) are shown in Figure 5A–C, respectively. The lower triacylglycerol content found in the liver of adult NR mice paralleled significantly decreased expression levels of *Srebf1* and *Plin2* in the liver of these animals as compared with controls. Trends towards decreased expression of *Mlxipl* (*p* = 0.081 for treatment effect, two-way ANOVA, mainly under HFD) and increased expression of *Ppara* (*p* = 0.057 for treatment effect, two-way ANOVA) were also detected in the liver of NR animals. These results may be suggestive of decreased lipogenesis/lipid droplet formation and increased FAO in the liver of NR mice in the fed state. Nevertheless, a trend in reduction of liver *Cpt1a* expression levels was apparent in the NR mice (*p* = 0.071 for treatment effect, two-way ANOVA). Liver *Ucp2* expression levels were increased after HFD in the control mice but not in the NR mice, although the NR mice tended to have increased basal levels than controls on the NFD (*p* = 0.075, *t*-Student). Hepatic gene expressions of *Fasn and Acaca,* encoding key enzymes for de novo synthesis of fatty acids in the liver, were suppressed in the HFD-fed animals regardless of the neonatal NR treatment, which had no effect on the expression of these two genes. Hepatic gene expressions of *Pnpla2* and *Pparg* were unaffected by neonatal NR treatment or the fat content of the diet (Figure 5A,B). Regarding the genes related to the status of SIRT1 and AMPK pathways (Figure 5C), compared to their controls, adult NR mice had significantly increased hepatic *Sirt1* mRNA expression levels regardless of the type of diet (as indicated by two-way ANOVA) and significantly increased *Prkaa2* mRNA expression levels under NFD (*p* < 0.05, *t*-Student). Further, hepatic gene expression levels of SIRT1 transcriptional targets analyzed were greater in the NR mice as compared with the control mice under HFD (case of *Gadd45a*) or NFD (case of *Sod2*) feeding conditions. 

## 4. Discussion

NR has been attracting a lot of attention in recent years owing to accumulated evidence of the beneficial effects of its contemporary administration against multiple metabolic complications and age-related disorders involving NAD^+^ decline in tissues [47]. The programming activity of NR supplementation in early-life stages has been much less studied, only recently, and so far mostly regarding long-term effects on adipose tissue features (our previous reports [27,28,30]) and cognitive/behavioral skills [29]. Using the same study design as in the present work, we previously reported sex-specific long-term effects of early life NR supplementation favoring better systemic responses to HFD feeding and thermogenic/oxidative capacities in WAT, specifically in the male offspring [27]. Here, we pursued a more comprehensive characterization of the programming activity of NR in the male progeny and reported novel results revealing long-term beneficial effects of direct NR supplementation to suckling mice on lipid and energy metabolism in SM and liver in adulthood. Results herein and in our previous works are noteworthy, considering the low NR dosage used and that treatment was limited to the suckling period, while the animals were challenged with an obesogenic diet and analyzed much later in life. Importantly, our study design allows studying direct programming effects of oral NR, independent of changes in milk composition that are inherent to (and demonstrated in) other designs, namely NR supplementation to the lactating dams [29]. 

Even if a non-obesity prone mouse strain was used in this work (NMRI mice), the HFD regimen applied (10 weeks duration, 45% calories as fat) did increase body weight, adiposity, and leptinemia (Appendix A and [27]), as well as (this work) muscle triacylglycerol content and the degree of liver steatosis in our control mice, although it failed to alter insulin sensitivity or glucose control [27]. The observed HFD-induced increase in muscle triacylglycerol content is in good concordance with previous reports of this effect in multiple mouse strains after just 8 weeks of an HFD similar to the one used here ([48] and references therein). The fact that the HFD-induced increase in liver steatosis—which is a histological diagnosis of fat accumulation—was dissociated from an increase in the liver triacylglycerol content is not without precedent. For instance, in studies examining the timeline of the onset of non-alcoholic fatty liver disease in HFD-fed (60% energy as fat), obesity-prone (C57BL/6J) male mice, histological steatosis appeared well before a persistent, significant increase in liver triacylglycerol content could be detected (week 4 versus week 7–8) [49]. 

Interestingly, male mice supplemented with NR during the suckling period were completely protected against HFD-induced triacylglycerol accumulation in SM and had lower triacylglycerol content and steatosis in the liver after both NFD and HFD feeding compared to their control siblings. From the muscle results, protection against HFD-induced insulin resistance/prediabetes can be anticipated in the mice neonatally treated with NR, considering the close relationship between muscle (intramyocellular) lipid content and systemic insulin resistance, and that muscle lipids primarily reflect muscle triacylglycerol content (even if other lipid species, such as diacylglycerol or ceramides, mediate the unwanted effects) [50]. Likewise, protection against HFD-induced fatty liver progression is suggested by the liver results. Protective effects of early-life NR supplementation against metabolic complications under more severe models or pro-obesogenic feeding conditions in adulthood deserve further investigation, although it might be that, under such severe conditions, mild programming effects of low-dose neonatal nutritional treatments are blunted.

Protection against HFD-induced triacylglycerol accumulation in SM in the NR mice could be related to gene expression changes in the tissue observed selectively in these animals, but not in controls, following HFD, such as the HFD-induced increases in *Plin5* and *Cpt1b* expression and decrease in *Acacb* expression. These changes in concert could favor increased dietary fat oxidation in muscle in the NR mice through increased Perilipin5-mediated enhancement of mitochondria function [41], increased CPT1 abundance, and increased CPT1 activity owing to lower levels of inhibitory malonyl-CoA. Furthermore, NR mice may be better prepared than controls to handle the excess dietary fat through oxidative metabolism in SM from the very beginning of the HFD challenge, as they displayed increased *Ucp3* gene expression, as well as a trend to increased mitochondria content under basal (NFD) conditions. In good concordance with increased FAO in SM are also results herein suggesting a relative activation of SIRT1 and AMPK pathways in SM of NR mice as compared to controls. Dissociation of AMPK phosphorylation from ACC phosphorylation in the context of muscle FAO control, as observed here in the NR mice, has been reported previously in some settings [51,52]. In this context, it is to be noted that, besides acute stimulatory effects on FAO linked to AMPK-catalyzed ACC phosphorylation, active AMPK favors FAO through the induction of the expression of FAO-related genes, such as *Cpt1b* and *Ucp3* [52,53,54]. The latter mechanisms appear to be preponderant in our experimental setting. Regarding the programming of liver metabolism, lower liver triacylglycerol content in the adult NR mice, as observed, could be related to increased FAO (up-regulated *Ucp2* and *Ppara* expressions in NFD) and, more consistently, decreased lipogenesis and lipid droplet formation (down-regulated *Srebf1*, *Mlxipl,* and *Plin2* expressions). As for muscle, results obtained are compatible with a relative activation of SIRT1 and AMPK pathways in the liver of adult NR mice as compared with their control siblings.

The study protocol used to assess long-term metabolic programming activity was conducted in parallel with early-life supplementation with NR (this work) and with the polyphenol resveratrol (RSV) [31]. NR and RSV have both been described as activators of SIRT1 [5,55,56], which is an important regulator of cellular energy metabolism [57] and a known epigenetic landscaper modulator [58]. In fact, early-life supplementation with NR (this work) or RSV [31] had similar effects on lipid metabolism in SM of male mice in adulthood and also effects in common in the liver. However, differences between NR and RSV programming effects were also noticed, especially regarding adult liver traits and responses. In particular, decreased liver triacylglycerol content and attenuation of liver steatosis were significant or apparent in the NR mice only. Differences were to be expected since NR and RSV have additional, non-overlapping biological targets for interaction besides SIRT1. In fact, the effects of the two neonatal treatments on WAT features in adulthood (including DNA methylation marks in browning-related genes in WAT) and the adipogenic fate of primary preadipocytes of young animals were similar but not identical [27,28,30]. For instance, long-term effects of early-life NR supplementation on WAT gene expression were seen mainly under HFD feeding conditions, which was not the case for early-life RSV supplementation [27].

The ultimate mechanisms explaining the distinct phenotypes observed in adulthood in neonatally NR-treated mice remain to be established. Strikingly, the early-life NR treatment applied had multiple effects on gene expression and responses to HFD feeding in SM and liver in adulthood, yet it did not affect the expression profile of studied genes in SM or liver of mice at a young age (35 days; Appendix A). There is evidence that NR treatment can impact the cellular DNA methylation machinery, at least in adipocytes/adipose tissue [30], and it is conceivable that supplemental NR during the suckling period (or a stimulus subsequent to NR supplementation, such as SIRT1 activation) triggers an epigenetic effect or wave of effects in SM and/or liver tissues that is revealed, in terms of gene expression, later in life. However, early-life NR treatment in male mice leads to enhanced responses of WAT and BAT oxidative metabolism to HFD feeding in adulthood [27], and therefore, it cannot be overlooked that changes observed in muscle and liver of neonatally NR-treated mice might be secondary to the programming of more oxidative adipose tissues. Clearly, much more needs to be investigated regarding the mechanisms behind the metabolic programming of muscle and liver traits linked to early-life NR supplementation revealed, for the first time to our knowledge, in this work. 

Taken together with our previous results [27], results herein suggest that, similar to early-life RSV supplementation, early-life NR supplementation could represent an adequate, multi-target-directed strategy to prevent obesity and metabolic disturbances related to ectopic fat accumulation in non-adipose tissues in adulthood in the male progeny. The sex-specificity of this recommendation should be stressed, particularly for NR, because our previous findings indicated not only a lack of induction of adult WAT browning but also alterations of concern in systemic glucose and lipid metabolism parameters in adulthood in the neonatally NR-supplemented female mice [27]. 

## Figures and Tables

**Figure 1 nutrients-14-02259-f001:**
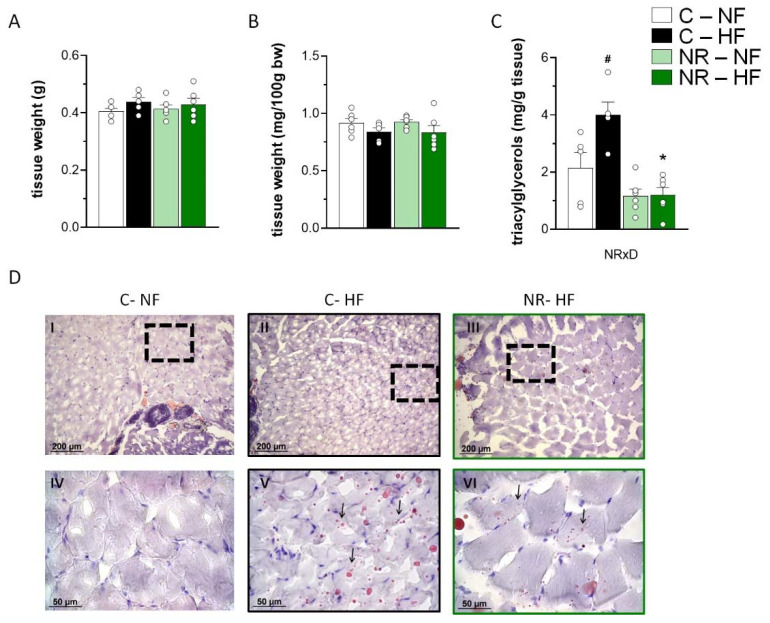
Early-life nicotinamide riboside (NR) treatment protected against high-fat (HF) diet-induced triacylglycerol accumulation in skeletal muscle in adulthood. Newborn male mice received NR or vehicle (water; control) from day 2 to 20 of age, were weaned onto a chow diet on day 21, and were assigned to either an HF or a normal-fat (NF) diet on day 90 for 10 weeks. *Gastrocnemius* muscle weight (**A**), weight as a percentage of body weight (**B**), and triacylglycerol content (**C**) at the end of the experiment. Data are the mean ± SEM of 5–6 animals per group. Circles correspond to individual data. Statistics (*p* < 0.05): NRxD treatment per diet interaction (two-way ANOVA); * NR vs. control, ^#^ HF vs. NF (*t*-Student). In (**D**), representative Oil Red O staining of *Gastrocnemius* muscle sections of indicated experimental groups are shown (panels I-III), together with corresponding magnification of indicated areas in boxes (panels IV-VI). Arrows in images denote intramyocellular neutral lipid droplets.

**Figure 2 nutrients-14-02259-f002:**
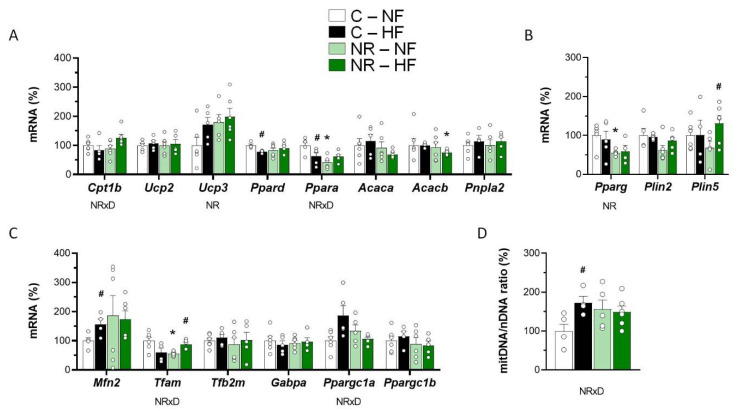
Early-life nicotinamide riboside (NR) treatment impacted lipid metabolism and mitochondria pathways in skeletal muscle in adulthood. Newborn male mice received NR or vehicle (water; control) from day 2 to 20 of age, were weaned onto a chow diet on day 21, and were assigned to either an HF or a normal-fat (NF) diet on day 90 for 10 weeks. mRNA levels of the indicated genes related to fatty acid oxidation/lipolysis (**A**), lipogenesis/lipid droplet (**B**), and mitochondria biogenesis and function (**C**) were analyzed in *Gastrocnemius* skeletal muscle at the end of the experiment. The mitochondrial DNA to nuclear DNA ratio of the same animals as an indicator of mitochondrial content is shown in (**D**). Data are the mean ± SEM of 5–6 animals per group and are expressed relative to the mean value of the NF-control group, which was set to 100. Circles correspond to individual data. Statistics (*p* < 0.05): NR treatment, and NRxD treatment per diet interaction (two-way ANOVA); * NR vs. control, ^#^ HF vs. NF (*t*-Student).

**Figure 3 nutrients-14-02259-f003:**
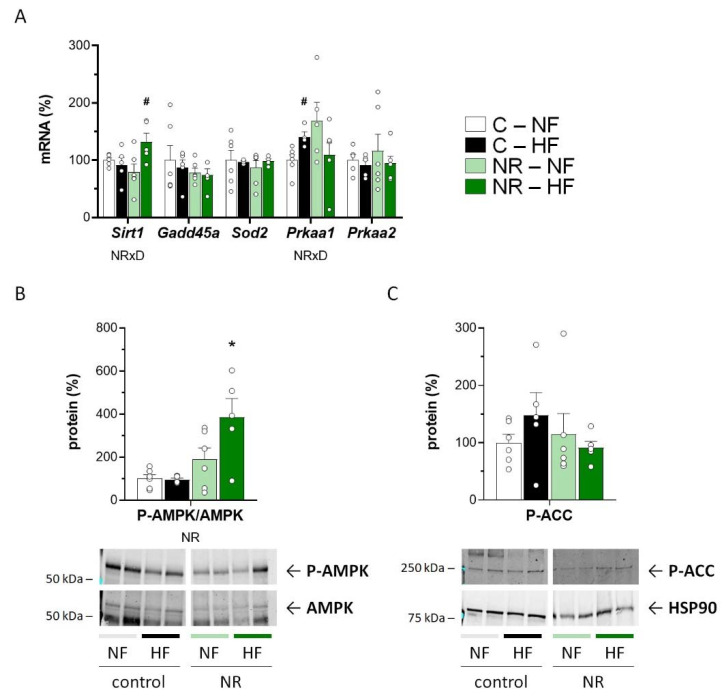
Early-life nicotinamide riboside (NR) treatment up-regulated AMPK and SIRT1 in skeletal muscle in adulthood. Newborn male mice received NR or vehicle (water; control) from day 2 to 20 of age, were weaned onto a chow diet on day 21, and were assigned to either an HF or a normal-fat (NF) diet on day 90 for 10 weeks. mRNA levels of the indicated genes related to SIRT1 and AMPK pathways (**A**), phospho-AMPK to AMPK ratio (**B**), and phospho-ACC protein levels (**C**) were analyzed in *Gastrocnemius* skeletal muscle at the end of the experiment. Data in the histograms are the mean ± SEM of 5–6 animals per group and are expressed relative to the mean value of the NF-control group, which was set to 100. Circles correspond to individual data. Illustrative Western blots are shown at the bottom of panels B and C (see also Appendix A). Statistics (*p* < 0.05): NR treatment, and NRxD treatment per diet interaction (Two-way ANOVA); * NR vs. control, ^#^ HF vs. NF (*t*-Student).

**Figure 4 nutrients-14-02259-f004:**
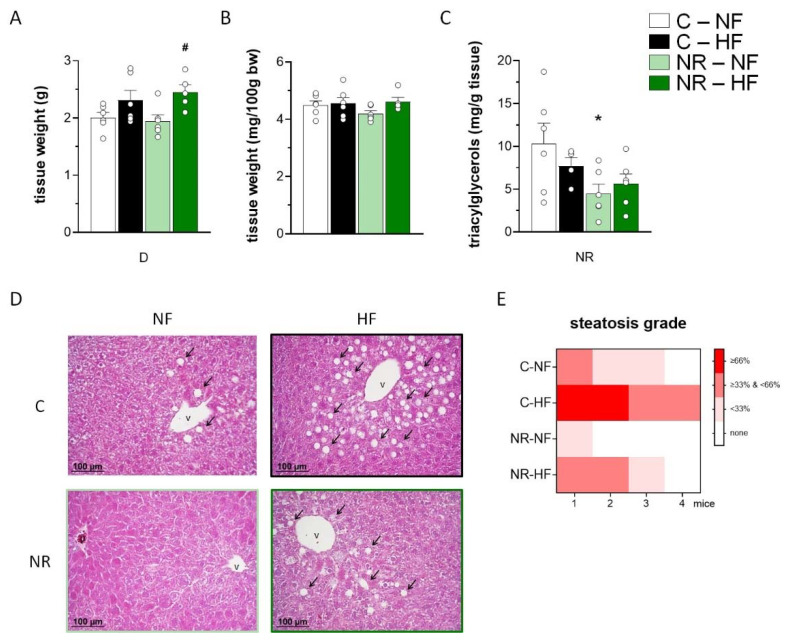
Early-life nicotinamide riboside (NR) treatment resulted in decreased liver triacylglycerol content in the liver in adulthood. Newborn male mice received NR or vehicle (water; control) from day 2 to 20 of age, were weaned onto a chow diet on day 21, and were assigned to either an HF or a normal-fat (NF) diet on day 90 for 10 weeks. Liver weight (**A**), weight as a percentage of body weight (**B**), and triacylglycerol content (**C**) at the end of the experiment. Data are the mean ± SEM of 5–6 animals per group. Circles correspond to individual data. Statistics (*p* < 0.05): NR treatment, and D diet effect (two-way ANOVA); * NR vs. control, ^#^ HF vs. NF (*t*-Student). Representative micrographs (magnification × 10) of H&E stained liver sections (**D**) and heat map of hepatic steatosis grade in liver biopsies from four mice (**E**) of each experimental group are shown. Symbol images: arrows denote macrovesicular steatosis, v indicates central vein, p indicates portal vein.

**Figure 5 nutrients-14-02259-f005:**
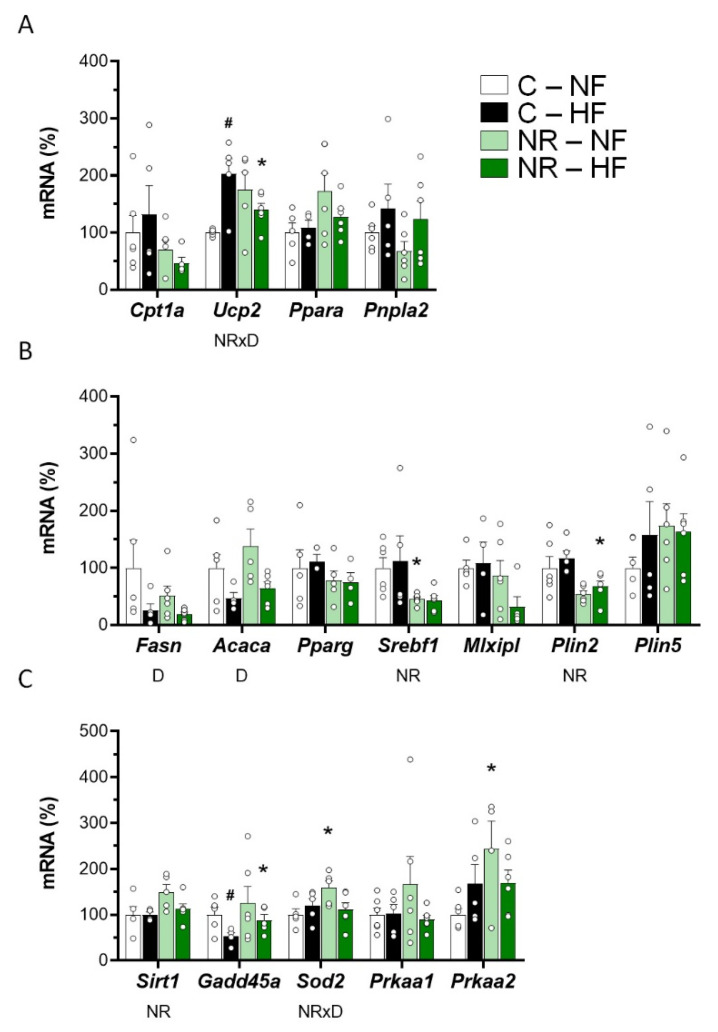
Early-life nicotinamide riboside (NR) treatment modified lipid metabolism capacities in the liver in adulthood. Newborn male mice received NR or vehicle (water; control) from day 2 to 20 of age, were weaned onto a chow diet on day 21, and were assigned to either an HF or a normal-fat (NF) diet on day 90 for 10 weeks. mRNA levels of the indicated genes related to fatty acid oxidation/lipolysis (**A**), lipogenesis/lipid droplet (**B**), and SIRT1 and AMPK pathways (**C**) were analyzed in the liver at the end of the experiment. Data are the mean ± SEM of 5–6 animals per group and are expressed relative to the mean value of the NF-control group, which was set to 100. Circles correspond to individual data. Statistics (*p* < 0.05): NR treatment, D diet, and NRxD treatment per diet interaction (two-way ANOVA); * NR vs. control, ^#^ HF vs. NF (*t*-Student).

## Data Availability

All data collected are reported in the manuscript and the Appendix A.

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
