# Peer review of "Nicotinamide Riboside Supplementation to Suckling Male Mice Improves Lipid and Energy Metabolism in Skeletal Muscle and Liver in Adulthood"

_nutrients, 2022, doi:10.3390/nu14112259_

Round 1
Reviewer 1 Report
In the present work, the authors study the impacts of early supplementation with nicotinamide riboside at the level of muscle and liver metabolism in adult mice. The current work is interesting and it is well written.
However, in my opinion, the authors should consider the revision of some aspects that could improve their manuscript.
As this is a continuation of previous studies, the authors have decided not to re-include data that already appeared in previous publications. However, I think that at least the final weight of the animals should be shown as supplementary material to help the reader.
In the text, the authors indicate: ‘To be noted, microscopic examination of hematoxylin and eosin stained liver sections suggested less hepatocellular ballooning in the NR mice as compared to the control mice (Figure 4D).’ This perception is not obvious to a person unfamiliar with such as term, and even less considering the small size/poor resolution of the images. Authors could zoom in, do some counting, or at least use arrowheads to better highlight what they mean. Otherwise, that claim is highly speculative.
In the Figure 3 (B and C), the authors presented some densitometry results of western blot analysis. However, the authors did not provide the blots, nor a representative image of these analyses. The authors should provide this information to give credibility to these results.
Author Response
In the present work, the authors study the impacts of early supplementation with nicotinamide riboside at the level of muscle and liver metabolism in adult mice. The current work is interesting and it is well written.
Thank you
As this is a continuation of previous studies, the authors have decided not to re-include data that already appeared in previous publications. However, I think that at least the final weight of the animals should be shown as supplementary material to help the reader.
A supplementary Table (Supplementary Table 1) is now included which shows final body weight, individual WAT depots weight and adiposity index, as well as energy intake and energy efficiency along the high fat diet challenge, for the four experimental groups.
In the text, the authors indicate: ‘To be noted, microscopic examination of hematoxylin and eosin stained liver sections suggested less hepatocellular ballooning in the NR mice as compared to the control mice (Figure 4D).’ This perception is not obvious to a person unfamiliar with such as term, and even less considering the small size/poor resolution of the images. Authors could zoom in, do some counting, or at least use arrowheads to better highlight what they mean. Otherwise, that claim is highly speculative.
We thank the referee very much in particular for this comment, as it allowed us to realize a conceptual mistake present in the original version, and to correct it in this revised version. The mistake is that we called “hepatocellular ballooning” to what is in fact “liver steatosis”, which is a histological diagnosis much easier to visualize than hepatocyte ballooning, and actually much more evident from the images shown in the Ms. We have now analyzed and semi-quantified the degree of steatosis in the histological liver sections of four animals in each group, according to the classification of Brunt et al. 1999 (doi:S0002927099004335; now reference 36 of the revised version). Results of this analysis are included in the revised Figure 4 (panel 4E), and they clearly indicated an increased degree of steatosis in the HFD-fed animals as compared to the NFD-fed animals, and a lower degree of steatosis in the mice treated with NR during the suckling period as compared to the control mice, both after NFD and HFD feeding in adulthood. We have introduced changes in the Abstract, subsection 2.6 of Materials and Methods, the description of Results (subsection 3.4), and the Discussion accordingly. We sincerely apologize for this mistake.
In the Figure 3 (B and C), the authors presented some densitometry results of western blot analysis. However, the authors did not provide the blots, nor a representative image of these analyses. The authors should provide this information to give credibility to these results.
Representative blots are now included in Figure 3, as requested.
Reviewer 2 Report
In this report Serrano et al measured the effects of Nicotinamide riboside supplementation on hepatic and muscle energy metabolism to suckling mice. They describe beneficial effects of NR on skeletal muscle and liver, where NR supplementation decreases hepatic and skeletal TG accumulation. They attempt to understand the mechanisms behind the improvements mostly by performing qPCR analysis of lipid metabolism related genes.
- While the concept is interesting and descriptive , the conclusions are speculative at best and no direct interrogation is provided for explanation of observed phenotypes.
- The authors should show the western blots of all the quantifications they provide in the paper, in conventional model, phosphorylation of AMPK should result in increase in pACC level.
- The authors should include individual data points in their graphs
- The authors should include the total body weight as well.
- What is the reason that there is no change in liver TG content after 10 weeks of HFD? What was the state of the sacrifice (fasted vs refed)? When describing hepatic lipogenic/lipolysis genes in liver it is imperative to mention the state of feeding. for example, mxlipl is highly regulated by feeding and it is unclear the feeding status of mice.
- While the study is interesting in design, the choice of mouse strain is not the best, as the paper would have benefited from steatosis prone strain.
Author Response
While the concept is interesting and descriptive, the conclusions are speculative at best and no direct interrogation is provided for explanation of observed phenotypes.
Thank you for finding the concept interesting. We agree that interrogation for explanation of observed phenotypes was rather confuse and insufficient in the original version. We have now devoted a paragraph of the Discussion, the penultimate one, to the mechanisms that may explain the distinct phenotypes observed in adulthood in the neonatally NR-treated mice, including the possibility of epigenetic effects and of indirect effects, secondary to the long-term effects of early-life NR treatment on oxidative metabolism in adipose tissues. While these mechanisms, as well as the translation potential of these pre-clinical results, remain speculative, we believe the conclusion that NR supplementation to suckling male mice improves lipid and energy metabolism in skeletal muscle and liver in adulthood is well supported by the results.
The authors should show the western blots of all the quantifications they provide in the paper, in conventional model, phosphorylation of AMPK should result in increase in pACC level.
Representative blots are now included in Figure 3, as requested. Indeed, in conventional model, phosphorylation of AMPK should result in an increase in pACC levels, but this was not indicated by western blot results in our experiment. Still, we considered important to include the data produced on the phosphorylation state of both proteins.
The authors should include individual data points in their graphs
Individual data points are now included in all histograms, as requested.
The authors should include the total body weight as well.
A supplementary Table (Supplementary Table 1) is now included which shows final body weight, individual WAT depots weight and adiposity index, as well as energy intake and energy efficiency along the high fat diet challenge, for the four experimental groups.
What is the reason that there is no change in liver TG content after 10 weeks of HFD?
Lack of increase of liver TG content after HFD could be due to the fact that we used a moderate HFD challenge (45% energy as fat) in a non-obesity prone strain (NMRI mice), and not, for instance, a 60% energy HFD in obesity prone C57BL/6J mice.
We emphasize (here and in the writing of the revised version) that, even if the HFD did not provoke an increase of the liver TG content, it led to increased liver steatosis, which is a histological diagnosis of fat accumulation. [NOTE: In the original version, unfortunately we mixed up steatosis with ballooning, by mistake, and this has now been corrected throughout the Ms, following the comment of one of the Reviewers].
The fact that the HFD increase in histological liver steatosis was dissociated from an increase in the liver TG content is not without precedent. For instance, in studies examining the timeline of the onset of non-alcoholic fatty liver disease in HFD (60% energy as fat)- fed obesity-prone C57BL/6J male mice, histological steatosis appeared well before a persistent significant increase in liver triacylglycerol content could be detected (week 4 versus week 7-8) (reference 49 of the revised version). This explanation is now included in the Discussion, second paragraph.
What was the state of the sacrifice (fasted vs refed)? When describing hepatic lipogenic/lipolysis genes in liver it is imperative to mention the state of feeding. for example, mxlipl is highly regulated by feeding and it is unclear the feeding status of mice.
The animals were euthanized under fed conditions, within the first 2h of the light cycle. This was (and it is) stated in subsection 2.1 of Materials and Methods, and it is now also remarked in the description of the liver results (subsection 3.4) of revised version, to stress this important point, and in view of the Reviewer comment.
While the study is interesting in design, the choice of mouse strain is not the best, as the paper would have benefited from steatosis prone strain.
Thank you for finding the study design interesting. Even if a non-obesity prone mouse strain was used (NMRI), the HFD regimen applied did increase body weight, adiposity and leptinemia (supplementary Table 1 and reference [27]), as well as (this work) muscle triacylglycerol content and the degree of liver steatosis especially in our control mice. This sentence is included in the revised version of the Ms (Discussion section, beginning of the second paragraph).
We agree with the Reviewer that protective effects of early-life NR supplementation against metabolic complications under more severe models or pro-obesogenic feeding conditions in adulthood deserve further investigation, although it might be that, under such severe conditions, mild programming effects of low-dose neonatal nutritional treatments are blunted. These considerations are also included in the revised version of the Ms (Discussion section, end of the third paragraph).
Reviewer 3 Report
This study attempts to assess whether early-life supplementation with NR in mice can impact adulthood lipid and energy metabolism, specifically in liver and skeletal muscle. The interest of this being a potential preventative strategy for obesity-related disorders characterised by ectopic lipid accumulation. The effects of both NR supplementation and high/normal fat diet intervention were studied.
The results shown are only for male mice, despite the study implying that female mice also undertook the NR intervention. It should be explained why the results of this are not shown, or they should be included. A sex-specific result on a previous NR study was mentioned within the discussion. Can the authors provide any insight into the possble reasons for this reported sex-specificity?
There were 5-6 animals/grp from 4-5 different litters. Were mice from the same litter split between the groups, or were they placed within the same interventional group?
Skeletal muscle TG determination - using the methods outlined, this would include both intramyocellular and extramyocellular TGs components. This should be mentioned.
Minor points:
- Abstract: ".... distinct capacities for fat oxidation and lipogenesis". Should this be 'reduced lipogenesis'?
- Introduction: BDNF acroymn in full
- Fig 1D. Dotted box needs making more bold (it is very hard to see).
- Although a reference to the HFD is made, it would be helpful to include %SFA,MUFA,PUFA within the methods too.
Author Response
The results shown are only for male mice, despite the study implying that female mice also undertook the NR intervention. It should be explained why the results of this are not shown, or they should be included. A sex-specific result on a previous NR study was mentioned within the discussion. Can the authors provide any insight into the possble reasons for this reported sex-specificity?
Indeed, females in the litters also received vehicle or NR during the suckling period, and they were monitored for a number of parameters. We focus in males because we previously reported sex-specific long-term effects of early life NR supplementation favoring better systemic responses to HFD feeding and thermogenic/oxidative capacities in WAT specifically in the male offspring, and not the female offspring [27], and in the present work we pursued a more comprehensive characterization of the programming effects of NR in the male progeny. This was indicated at the end of the Introduction, and it is now re-stated in the first paragraph of the Discussion of the revised version as well, in view of the Reviewer’s comment. Also, in view of this comment, we have changed the title to include the sex of the animals, and avoid any misleading. We agree the sex-specificity of NR programming effects and its mechanisms deserve further research, yet these aspects are outside the direct scope of this work.
There were 5-6 animals/grp from 4-5 different litters. Were mice from the same litter split between the groups, or were they placed within the same interventional group?
Mice in each litter were split among the four study groups, and this is now explicitly stated in the revised version (Materials and Methods, subsection 2.1).
Skeletal muscle TG determination - using the methods outlined, this would include both intramyocellular and extramyocellular TGs components. This should be mentioned.
This is now mentioned in the description of the corresponding results (subsection 3.1)
Minor points:
Abstract: ".... distinct capacities for fat oxidation and lipogenesis". Should this be 'reduced lipogenesis'?
Reduced lipogenesis is now used in the abstract.
Introduction: BDNF acronym in full.
BDNF acronym in full is now used.
Fig 1D. Dotted box needs making more bold (it is very hard to see).
The dotted box in this Figure are now bolder.
Although a reference to the HFD is made, it would be helpful to include %SFA,MUFA,PUFA within the methods too.
The fact that fat in this commercial HFD comes mainly from lard (saturated fat) is now included in Methods.
Reviewer 4 Report
The manuscript by Alba Serrano and coll. reports the effect of early administration of nicotinamide riboside (NR) on lipid metabolism and lipid accumulation in liver and adipose tissues, in mice. Moreover, the effect of high and low-fat diets was compared in NR-treated mice. Although the manuscript is interesting, the results are not convincing.
The authors stated that they administered approximately 15 times more NR than the amount of Vit B3 ingested daily in milk. But what is the real amount? Authors should report the actual amount adsorbed by the animals (a measure of the plasma amount?).
Additionally, the authors stated that NR influenced muscle triacylglycerol levels, especially in HF mice. But this is not evident from the figure, where NR appears to have an effect in both NF and HF groups. Why is the NR-NF histological data not shown in Figure 1D?
Furthermore, authors often describe differences that are not evident from the figures either with the values ​​or with the asterisks.
Finally, it is not clear why C-HF mice have a lower hepatic triacylglycerol level than C-NF, although almost all literature has shown the opposite.
Author Response
The manuscript by Alba Serrano and coll. reports the effect of early administration of nicotinamide riboside (NR) on lipid metabolism and lipid accumulation in liver and adipose tissues, in mice. Moreover, the effect of high and low-fat diets was compared in NR-treated mice. Although the manuscript is interesting, the results are not convincing.
We are glad the Reviewer found the Ms interesting, and hope that, in this revised version, he/she finds the results convincing.
The authors stated that they administered approximately 15 times more NR than the amount of Vit B3 ingested daily in milk. But what is the real amount? Authors should report the actual amount adsorbed by the animals (a measure of the plasma amount?).
As indicated in our previous study (reference 27), the 15x dose was an estimation, based on B3 vitamin levels in milk of mice dams in our animal house and previously reported average daily milk intake in rodents. Our aim was to provide a mild extra dose of B3 as NR. We agree that measurements of plasma B3 forms in the vehicle and NR-treated weanlings (right after the treatment period was finished) would strengthen the Ms. Unfortunately, we lack plasma of the 21-day-old weanlings. Nevertheless, we want to emphasize that all results in the Ms (except for those in supplementary figures 1 and 2) refer to the animals in adulthood (almost 5.5-month-old), when differences in plasma levels of B3 forms are not expected.
Additionally, the authors stated that NR influenced muscle triacylglycerol levels, especially in HF mice. But this is not evident from the figure, where NR appears to have an effect in both NF and HF groups. Why is the NR-NF histological data not shown in Figure 1D?
Early-life NR treatment had an effect on adult muscle triacylglycerol (TG) levels after HFD only, as concluded from the Figure 1C statistics: a significant NRxD effect in two-way ANOVA reflecting that HFD increased muscle TG content in the control mice but not the NR mice, and significantly lower muscle TG levels than controls in the NR mice after HFD (but not after NFD), according to t-Student test. The NR-NF was not processed for Oil Red staining in view of the triacylglycerol results in Figure 1C. Figure 1D includes representative Oil-Red O staining of histological sections confirming and illustrating the two main conclusions in Figure 1C, namely, increased muscle TG content in the C-HF mice as compared to the C-NF mice, and lower TG content in the NR-HF mice as compared to the C-HF mice.
Furthermore, authors often describe differences that are not evident from the figures either with the values ​​or with the asterisks.
We have changed the redaction of the Results section to put more focus on significant effects (p<0.05). We highlight trends when 0.05<p<0.1 (this was already and is stated in subsection 2.7 of Materials and Methods) and the trends are in line with the scenario indicated by the significant results.
Finally, it is not clear why C-HF mice have a lower hepatic triacylglycerol level than C-NF, although almost all literature has shown the opposite.
The hepatic triacylglycerol (TG) content was not different in the C-NF and the C-HF group. Lack of increase in hepatic TG content after HFD could be due to the fact that we used a moderate HFD challenge (45% energy as fat) in a non-obesity prone strain (NMRI mice), and not, for instance, a 60% energy HFD in obesity prone C57BL/6J mice.
In any case, we emphasize (here and in the writing revised version) that, even if the HFD did not provoke an increase of the liver TG content, it led to increased liver steatosis, which is a histological diagnosis of fat accumulation. [NOTE: In the original version, we mixed up steatosis with ballooning by mistake, and this has now been corrected throughout the Ms, following the comment of one of the Reviewers].
The fact that the HFD-induced increase in histological liver steatosis was dissociated from an increase in the liver TG content is not without precedent. For instance, in studies examining the timeline of the onset of non-alcoholic fatty liver disease in HFD (60% energy as fat)- fed obesity-prone C57BL/6J male mice, histological steatosis appeared well before a persistent significant increase in liver triacylglycerol content could be detected (week 4 versus week 7-8) (reference 49 of the revised version). This explanation is now included in the Discussion, second paragraph.
Round 2
Reviewer 1 Report
The authors have successfully incorporated all my comments and have improved the manuscript.
Author Response
Thank you.
Reviewer 4 Report
Authors have responded convincingly to my remarks
Author Response
Thank you.